# LOTUS: Latent Outpainting Diffusion Model for Three-Dimensional Ultrasound Stitching

**Xing Yao**[1]                                                            XING.YAO@VANDERBILT.EDU

**Runxuan Yu**[1]                                                       RUNXUAN.YU@VANDERBILT.EDU

**Nick DiSanto**[1]                                               NICOLAS.C.DISANTO@VANDERBILT.EDU

**Ehsan Khodapanah Aghdam**[1]                   EHSAN.KHODAPANAH.AGHDAM@VANDERBILT.EDU

**Kanyifeechukwu Oguine**[1]                   KANYIFEECHUKWU.J.OGUINE@VANDERBILT.EDU

**Daiwei Lu**[1]                                                          DAIWEI.LU@VANDERBILT.EDU

**Ange Lou**[1]                                                           ANGE.LOU@VANDERBILT.EDU

**Jiacheng Wang**[1]                                          JIACHENG.WANG.1@VANDERBILT.EDU

**Dewei Hu**[2]                                                               HU.DEWEI@MAYO.EDU

**Gabriel Arenas**[3]                                 GABRIEL.ARENAS@PENNMEDICINE.UPENN.EDU

**Baris Oguz**[3]                                                           BARISUMOG@GMAIL.COM

**Alison Pouch**[3]                                             POUCH@PENNMEDICINE.UPENN.EDU

**Nadav Schwartz**[3]                               NADAV.SCHWARTZ@PENNMEDICINE.UPENN.EDU

**Brett C Byram**[1]                                              BRETT.C.BYRAM@VANDERBILT.EDU

**Ipek Oguz**[1]                                                       IPEK.OGUZ@VANDERBILT.EDU

[1] *Vanderbilt University*

[2] *Mayo Clinic*

[3] *University of Pennsylvania*

**Editors:** Accepted for publication at MIDL 2025

## Abstract

3D ultrasound (3DUS) stitching can enlarge the field-of-view (FOV) by registering partially overlapping 3DUS images collected from different probe positions. However, standard registration algorithms frequently encounter difficulties with this task, primarily due to the sector-shaped FOV, which often leads to pronounced local minima, thereby obstructing optimization efforts. To address these limitations, we propose LOTUS, a novel Latent Diffusion Model (LDM) specifically designed for 3DUS FOV outpainting. LOTUS innovatively encodes the 3DUS data into a compact latent space and performs outpainting at test time, effectively extending the sector-shaped FOV into a standard rectangular shape. This transformation facilitates a more robust registration by mitigating the issues of local minima associated with the original FOV shape. Experimental results show that LOTUS significantly improves the accuracy of the registration as well as the efficiency of the outpainting process compared to existing models. The code is available at github.com/MedICL-VU/LOTUS.

**Keywords:** Latent Diffusion Model, Ultrasound, Outpainting, Registration

## 1. Introduction

US image registration (Che et al., 2017; Entrekin et al., 2001; Wang et al., 2014) is a pivotal task for downstream US analysis. A particularly important application is image stitching (Banerjee et al., 2015; Gomez et al., 2019; Wright et al., 2023; Bano and Stoyanov, 2024),

which can compound multiple US images collected from different probe positions by aligning the overlapping image contents to extend the US field of view (FOV). This is important for complete visualization of larger anatomical structures, such as the fetus and the placenta during the second/third trimesters of pregnancy (Roy-Lacroix et al., 2017; Gomez et al., 2017). However, the sector-shaped FOV inherent to US imaging poses considerable challenges for effective image stitching (Yao et al., 2024). It introduces a strong trivial local minimum of the similarity metric at the initial stage, rendering registration optimization difficult. Limiting the metric computation to just the overlapping region can help this problem, but this introduces a further complication as the similarity metric within the overlapping region can be trivially optimized by artificially reducing overlap. In this study, we explore the potential of outpainting the 3DUS volume to obtain a rectangular FOV for alleviating these optimization problems during registration.

Image outpainting has become a prominent topic in computer vision in recent years, with diffusion model (DM)-based methods achieving remarkable performance in natural image outpainting (Avrahami et al., 2023; Lugmayr et al., 2022; Corneanu et al., 2024; Ju et al., 2024; Xie et al., 2023; Zhuang et al., 2025). However, all these methods face challenges with artifacts between the original and synthetic regions, and they are primarily restricted to 2D domains, without the ability to maintain consistency across slices in a 3D volume. In the context of US outpainting, echoGAN (Gazda et al., 2024) introduced a conditional GAN to outpaint 2D cardiac US images from a smaller-angle FOV to a larger-angle FOV while maintaining the sector shape. SynStitch (Yao et al., 2024) proposed a ControlNet-based framework for 2D kidney US image outpainting. Nevertheless, both echoGAN and SynStitch focus on 2DUS and are unable to outpaint to a rectangular FOV. Outpainting for US images, and particularly for 3DUS, thus remains underexplored.

Unlike 3D outpainting task on the other medical imaging modalities (Liman et al., 2024; Li et al., 2024) with rectangular FOVs, outpaint a sector-shape FOV 3DUS to a rectangular FOV presents unique challenges. These challenges stem from the absence of whole-FOV ground truth to train the outpainting network in a supervised manner. Training models on sector-shaped FOV images restricts outpainting capabilities, as the model tends to adopt the sector shape as prior knowledge. Furthermore, the sector-shaped FOV introduces substantial missing regions, further increasing the complexity of 3DUS outpainting.

To address these challenges, we propose LOTUS: a Latent Diffusion Model (LDM) (Rombach et al., 2022) specifically designed for 3DUS FOV outpainting, inspired by the previous work on 2D natural images (Lugmayr et al., 2022; Corneanu et al., 2024).

- To the best of our knowledge, LOTUS is the first method to address the challenging task of outpainting the sector-shaped FOV of 3DUS into a rectangular shape. LOTUS executes 3D outpainting in the latent space instead of the image space at inference time. It can achieve realistic results while dramatically improving the inference speed with lower computational burden compared to its image-space counterparts.

- We propose an effective strategy to extract rectangular FOV patches to train the outpainting LDM, introducing rectangular FOVs as a prior knowledge to the model.

- To address inconsistencies between the outpainted regions and the original image, we propose a Latent Mask Generator (LMG) that preserves the majority of content-

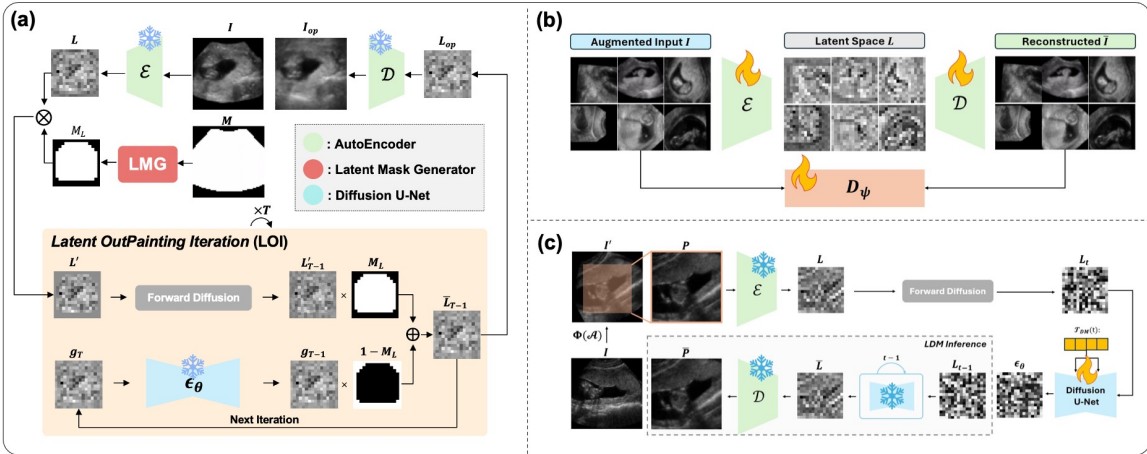

Figure 1: (a) Test-time latent space outpainting with LOTUS. (b) The training of AutoEn-coderKL. $\mathcal{E}$, $\mathcal{D}$, and $D_\psi$ are encoder, decoder, and patch-based discriminator, respectively. (c) The training and inference of LDM.

related features while effectively excluding FOV-edge-related features to achieve seam-less outpainting between the original and synthetic regions.

- We propose improving 3DUS image registration performance using FOV-outpainted 3DUS images, and we show this significantly enhances registration performance.

## 2. Methods

LOTUS (Fig. 1(a)) performs outpainting during inference using a pretrained latent diffusion model (LDM), which combines an AutoEncoder (AE) with a Diffusion U-Net.

### 2.1. Training of AutoEncoderKL

As the first step of LOTUS, we train an AutoEncoderKL $\{\mathcal{E}, \mathcal{D}\}$, where the encoder $\mathcal{E}$ compresses the 3DUS image $I$ of size $N^3$ into a latent representation $L$ of size $(\frac{N}{S})^3$, with a scaling factor $S = 4$. The decoder $\mathcal{D}$ reconstructs $L$ back to the image space, producing $\bar{I}$. As depicted in Fig. 1(b), the spatial correspondence between $I$ and $L$ enables effective pixel-space outpainting through operations performed in the latent space. To enhance the AE's generalization, we apply random affine transformations and cropping to the input 3DUS images, generating variations in size and shape. Training follows the implementation of (Rombach et al., 2022) and employs an adversarial framework, where a patch-based discriminator $D_\psi$ learns to distinguish between the original $I$ and the reconstructed $\bar{I}$.

### 2.2. Training and Inference of the Latent Diffusion Model

After obtaining the pretrained AutoEncoder, we train the LDM to unconditionally generate realistic 3DUS images with a rectangular FOV. However, 3DUS images acquired with sector

array probes inherently exhibit sector-shaped FOVs. Training directly on these images restricts outpainting capabilities, as the model learns the sector shape as prior knowledge.

To address this issue, we employ augmentation and central patch extraction, as illustrated in Fig. 1(c). The input images are upsampled by half, such that they contain $(2N)^3$ voxels. A random affine transformation $\mathcal{A}$ is applied to the input image $I$, producing $I' = I(\Phi(\mathcal{A}))$. Next, a central patch $P$ of size $N^3$ is extracted and used for LDM training. During training, the pretrained encoder $\mathcal{E}$ encodes the extracted rectangular FOV patch $P$ into its latent representation $L$. In the forward diffusion process, a noisy latent feature $L_t$ is generated by iteratively adding Gaussian noise to $L$, computed using the closed-form expression: $L_t = \sqrt{\bar{\alpha}_t}L + \sqrt{1 - \bar{\alpha}_t}\epsilon$, where $\bar{\alpha}_t = \prod_{s=1}^{t} \alpha_s$ is the cumulative product of noise scheduling coefficients $\alpha_t$, and $\epsilon \sim \mathcal{N}(0, L)$ is Gaussian noise. In the denoising stage, the model $\epsilon_\theta$ predicts and removes the noise $\epsilon$ from $L_t$ at each step $t$, by minimizing the loss:

$$\mathcal{L}_{\mathcal{LDM}} = \mathbb{E}_{L,\epsilon \sim \mathcal{N}(0,L),t} \left[ \|\epsilon - \epsilon_\theta(L_t, t)\|^2 \right]. \tag{1}$$

In Fig. 1(c), the gray dashed box illustrates the iterative denoising process applied to the noisy latent representation. At each time step $t$, the latent representation is updated as $L_{t-1} = L_t - \epsilon_\theta(L_t, t)$, where $\epsilon_\theta(L_t, t)$ is the noise predicted by the frozen Diffusion U-Net. This noise is iteratively removed, progressively refining the latent representation until the clean latent state $\bar{L}$ is obtained. Finally, $\bar{L}$ is passed through the decoder $\mathcal{D}$ to reconstruct the 3DUS image $\bar{P}$ in pixel space.

### 2.3. Test Time Latent Space Out-Painting with LOTUS

As illustrated in Fig. 1(a), once the pretrained LDM is obtained, LOTUS performs outpainting in the latent space without additional training. Given an input 3DUS image $I$, its foreground mask $M$ is extracted using a thresholding function. The image $I$ is then encoded by the pretrained encoder $\mathcal{E}$ into its latent representation $L$, while the latent mask $M_L$ is derived from the binary mask $M$ using LMG.

**Latent Mask Generator (LMG):** The LMG generates a latent mask $M_L$ that preserves content-related features while excluding those associated with the sector-shaped FOV edges. This process consists of two steps: (1) The binary mask $M$ of size $N^3$ is downscaled to $(\frac{N}{S} - d)^3$, where $d$ is a small integer satisfying $d < \frac{N}{S}$. We use $d = 2$ in this work. This slight shrinking ensures that $M_L$ effectively filters out edge-related features while retaining essential content. (2) The downscaled mask is zero-padded to $(\frac{N}{S})^3$.

**Latent Outpainting Iteration (LOI):** The LOI begins by extracting the latent condition $L' = L \times M_L$, where the latent mask $M_L$ ensures that $L'$ retains only content-relevant regions while excluding sector-shaped edges. The objective of latent outpainting is to preserve the masked content in $L'$ while generating new latent features outside the mask. We refer to the foreground of $M_L$ as the "Condition Region of Interest (CROI)" and the background as the "Outpainting Region of Interest (OROI)" in the following discussion.

In the first iteration at time step $T$, $T-1$ steps of Gaussian noise are added to the latent condition $L'$, yielding a noisy latent condition $L'_{T-1}$. Simultaneously, a random Gaussian noise sample $g_T \sim \mathcal{N}(0, L)$ is processed by the pretrained Diffusion U-Net to produce a one-step denoised synthetic latent feature $g_{T-1} = g_T - \epsilon_\theta(g_T, T)$. Next, the CROI of the noisy latent condition $L'_{T-1}$ is merged with the OROI of the denoised synthetic latent

feature $g_{T-1}$, forming the one-step denoised latent feature $\bar{L}_{T-1}$, computed as: $\bar{L}_{T-1} = M_L \times L'_{T-1} + (1 - M_L) \times g_{T-1}$ (Fig. 1(a)).

In subsequent iterations, $g_T$ is updated using the one-step denoised latent feature $\bar{L}_{T-1}$ from the previous step. By integrating information from the CROI of $L'$ and the OROI of $g_{T-1}$, $\bar{L}_{T-1}$ guides the reverse diffusion process while preserving the latent condition $L'$. This ensures that $L'$ remains unchanged while guiding the outpainting within the OROI to follow a realistic distribution consistent with the latent condition.

After $T$ iterations of latent reverse diffusion, the final outpainted latent feature $L_{op} = \bar{L}_0$ is obtained. Finally, the pretrained decoder reconstructs the outpainted image $I_{op} = \mathcal{D}(L_{op})$.

## 2.4. Datasets and Implementation Details

We use 2 in-house placenta 3DUS datasets, **GenUS** for LOTUS outpainting training and testing, and **RegUS**, for registration testing. All data was acquired with a GE Voluson E8.
**GenUS Dataset:** This dataset comprises 99 3DUS placenta images from first-trimester subjects, where the placenta typically fits within a single 3DUS volume. All volumes are resampled to a spatial resolution of $(0.5\text{mm})^3$, centrally cropped to $(256)^3$ voxels, and intensity normalized to $[0, 1]$. The dataset is split 89:10 for LOTUS outpainting training and testing. During AutoEncoderKL training, the images are further resampled to $(4\text{mm})^3$, centrally cropped to $(64)^3$ voxels with zero-padding. We extract the foreground mask $M$ defined as the non-zero voxels. Extensive augmentation is applied, including random translations $([-12, 12])$, rotations $([-\frac{\pi}{2}, \frac{\pi}{2}])$, scaling $([0.8, 1.2])$, and random patch cropping (patches that don't overlap with $M$ are not allowed), each with a probability of 0.5. For Diffusion U-Net training, similar random affine augmentations are applied, followed by the extraction of a $(64)^3$ central patch with a rectangular FOV.
**RegUS Dataset:** The RegUS dataset is used to evaluate registration performance and consists of two 3DUS placenta volumes from each of 20 first-trimester subjects. All volumes are resampled to a spatial resolution of $(2\text{mm})^3$, centrally cropped to $(64)^3$ voxels, and intensity normalized to $[0, 1]$. The 'ground-truth' rigid transformations are manually annotated by two experts and further validated visually by three additional experts. This ground truth is used as an independent standard for evaluation. The manual registrations show the maximum rotation angles range between $[30°, 117°]$ and maximum translations between $[25, 83]$ mm. For registration, each subject's 3DUS volumes are registered bidirectionally (both A to B and B to A), yielding $2 \times 20 = 40$ registration pairs.

The data is empirically split into two categories, typical cases and hard cases. Hard cases consist of 20 pairs where there is, for example, a very large (e.g., $> 80°$) rotation along one axis, a very large translation (e.g., $> 60\text{mm}$), or image quality issues (e.g., shadow artifacts). These hard cases are a challenge for all compared methods (baseline and LOTUS). We hypothesize that a rough initial registration would help overcome registration optimization issues in these cases. We thus introduce a fixed initial rotation along the primary axis of rotation for each subject, which we call the Compensation Rotation (CR). We compare all methods for CR settings of $10°$, $20°$, or $30°$ to test our hypothesis.
**Outpainting Evaluation:** We compare LOTUS with RePaint (Lugmayr et al., 2022) and LOTUS*. RePaint is a test-time outpainting diffusion model originally designed for 2D natural images, which we extend to 3D for fair comparison. LOTUS* is a variant of

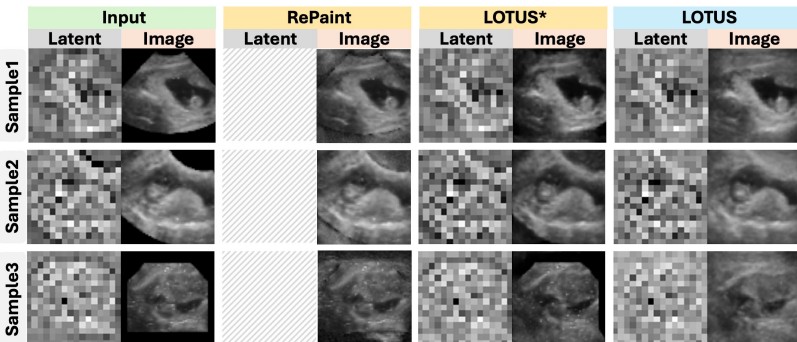

Figure 2: Typical outpainting results. The pairs of columns present the latent features and the corresponding images for each method. LOTUS effectively achieves a rectangular FOV, whereas RePaint contains an obvious seam surround the FOV edges, and LOTUS* fails to outpaint the blank areas.

LOTUS where the LMG module is replaced with a simple downsampling process (Corneanu et al., 2024). Outpainting performance is assessed using image similarity metrics, namely, normalized cross-correlation (NCC), structural similarity index measure (SSIM), and mean squared error (MSE). We also compare inference time and model parameter size.

**Registration Evaluation:** We register the pairs of images from the RegUS dataset using two of the most widely used conventional registration methods, Greedy (Yushkevich et al., 2016) and ANTs (Avants et al., 2008). We compare the performance of these algorithms using either the original sector-shaped FOV images or the LOTUS outpainting results. We use MSE similarity metric and rigid transformations for both methods. For LOTUS, given a fixed image $I_F$ and a moving image $I_M$, we first apply LOTUS to outpaint their FOVs, obtaining $I_F^{op}$ and $I_M^{op}$. We then rigidly register $I_F^{op}$ and $I_M^{op}$, and apply the resulting transform to the original moving image $I_M$. We report the same image similarity metrics, along with peak signal-to-noise ratio (PSNR), mean rotation error RE (L1 norm, degrees), and mean translation error TE (L2 norm, mm).

**Implementation Details:** We implemented RePaint (3D) and LOTUS using MONAI (Pinaya et al., 2023; Cardoso et al., 2022). The 3D diffusion model (DM) in RePaint employs a 3-level U-Net with 256, 256, and 512 encoder channels. Both AutoEncoderKL and Diffusion U-Net in LOTUS use a 3-level U-Net with channels of 32, 64, and 64. Training is conducted for 2000, 1500, and 200 epochs for the DM, AutoEncoderKL, and LDM, respectively, with a batch size of 1. Inference is conducted using DDPM (Ho et al., 2020) with 1000 sampling steps. All experiments are performed on an NVIDIA A6000 GPU.

## 3. Results and Discussion

### 3.1. Outpainting Results

Fig. 2 presents a qualitative comparison of FOV outpainting performance. For LOTUS and LOTUS*, results are shown in both latent and image spaces, whereas for RePaint, only image space results are available. Across all samples, LOTUS consistently produces

Table 1: Quantitative outpainting results. **Bold**: best, *significant (paired t-test, p<0.05). The inference time is reported for a batch size of 4 and inference step of 1000. LOTUS significantly outperforms RePaint for all outpainting window sizes (OW).

| Metric | RePaint | | | LOTUS | | |
|---|---|---|---|---|---|---|
| | OW=32 | OW=40 | OW=48 | OW=32 | OW=40 | OW=48 |
| NCC↑ | 0.522±0.082 | 0.600±0.052 | 0.779±0.048 | **0.713±0.095*** | **0.810±0.108*** | **0.920±0.037*** |
| SSIM↑ | 0.175±0.055 | 0.332±0.061 | 0.635±0.065 | **0.472±0.052*** | **0.613±0.048*** | **0.816±0.038*** |
| MSE(×10)↓ | 0.779±0.161 | 0.573±0.130 | 0.266±0.080 | **0.257±0.006*** | **0.209±0.005*** | **0.128±0.003*** |
| Param Size↓ | 759.4MB(3D DM) | | | **9.2MB(3D AE)+12.0MB(3D LDM)** | | |
| Inf Time↓ | 40 minutes/batch | | | **59 seconds/batch** | | |

high-quality outpainting results with a rectangular FOV. The generated images preserve the original structure and contrast while realistically filling blank regions with ultrasound-like textures. The results further demonstrate LOTUS's ability to handle diverse FOV orientations and sizes. In contrast, LOTUS*, lacking the LMG module, struggles to remove FOV edge-related features, leading to outpainting failures. RePaint has sub-optimal performance due to the challenges of 3D outpainting directly in pixel space, especially when the window size is small.

Table 1 presents the quantitative results on the **GenUS** validation dataset. Performance improves for both methods as the outpainting window size (OW) increases. We observe that LOTUS significantly outperforms RePaint across all metrics and scales, with a substantial advantage in inference time and memory efficiency. Appendix Fig. A1 illustrates LOTUS performance qualitatively with respect to the outpaint window size. LOTUS consistently generates realistic structures and details even when conditioned on very limited patch sizes.

### 3.2. US Image Registration Results

**Evaluation on Typical Cases:** Fig. 3 compares 3DUS registration performance with and without LOTUS outpainting. Applying Greedy and ANTs directly to sector-shaped FOV images results in poor registration performance, often just returning the identity matrix. In contrast, using LOTUS outpainting as input, both Greedy and ANTs consistently achieve

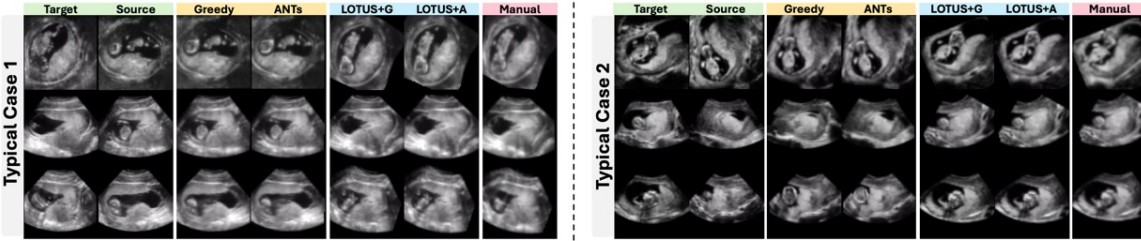

Figure 3: 3DUS registration on typical cases. For each subject, the rows show the axial, sagittal, and coronal planes, respectively. Greedy and ANTs fail on both cases. LOTUS achieves good performance by mitigating the influence of the FOV shape.

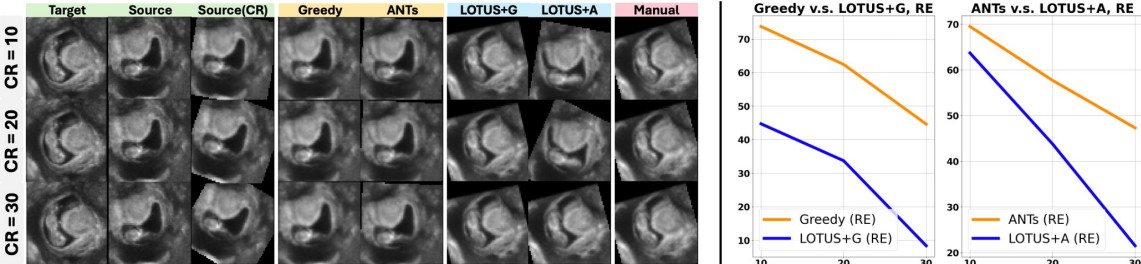

Figure 4: **Left**, Greedy and ANTs both fail on this randomly selected hard case, even with CR=30°. Interestingly, they even *reverse* the CR initialization as the optimization is dominated by FOV shape. In contrast, LOTUS leverages the initial CR to achieve good registration results. All panels show a center axial slice. **Right**, RE as a function of CR on hard cases. LOTUS significantly improves registration compared to the baselines.

accurate registration, closely aligning with manual registration. These findings highlight the effectiveness of LOTUS's FOV outpainting in enhancing 3DUS image registration.

We quantitatively assess performance by comparing the registration results to (a) the fixed image within the overlapping region, and (b) the manual registration ground truth (GT). While comparison to the fixed image mitigates any error in the manual GT, it may not fully reflect registration accuracy due to differences in contrast, artifacts, and deformation between the images. We thus report both in Table 2 to provide a comprehensive evaluation.

In Table 2-left, the moving image, as expected, exhibits the poorest similarity to the fixed image, serving as the lower bound. Both Greedy and ANTs show moderate improvements. In contrast, the proposed LOTUS-based methods significantly outperform their baselines, with LOTUS(ANTs) surpassing LOTUS(Greedy), suggesting a more robust alignment. Interestingly, LOTUS(ANTs) even outperforms the manual GT.

Table 2-right presents the registration results compared to the manual GT. We note that RE provides a direct measure of registration accuracy against the manual transformations, unlike the more indirect image similarity measures. The results are consistent with the left panel. Additional metrics (PSNR, SSIM, TE) are available in Appendix Tables A1 and A2.

Table 2: Registration performance on typical cases using the fixed image (left) and GT (right) for evaluation. **Best** and second best are highlighted. LOTUS outpainting significantly boosts performance (*, paired t-tests, p<0.05) compared to baselines.

| Method | Similarity Metrics | |
|---|---|---|
| | NCC ↑ | MSE(×10)↓ |
| MOV | 0.552±0.128 | 0.325±0.113 |
| GT | 0.892±0.033 | 0.127±0.034 |
| Greedy | 0.702±0.214 | 0.240±0.136 |
| ANTs | 0.698±0.157 | 0.237±0.102 |
| LOTUS+Greedy | 0.810±0.213 | 0.138±0.066* |
| LOTUS+ANTs | **0.891±0.053*** | **0.122±0.035*** |

| Method | Similarity Metrics | | Transformation Error |
|---|---|---|---|
| | NCC ↑ | MSE(×100) ↓ | RE ↓ |
| Greedy | 0.839±0.219 | 1.44±1.42 | 43.4±34.8 |
| ANTs | 0.851±0.113 | 1.49±0.95 | 55.1±33.8 |
| LOTUS+Greedy | 0.887±0.298 | 0.928±1.858 | 17.3±26.8* |
| LOTUS+ANTs | **0.985±0.023*** | **0.255±0.210*** | **8.04±17.67*** |

**Evaluation on Hard Cases:** We evaluate whether an initial transform helps performance in hard cases. Fig. 4-left shows a random hard case for CR values of 10°, 20°, and 30°. Both baseline methods fail due to the strong local minima induced by the sector-shaped FOV. These methods even reverse the initial CR back to the identity transform. In contrast, LOTUS-based methods leverage the CR initialization to refine registration and successfully align the images at CR = 30°, with LOTUS(Greedy) achieving success even at CR = 10°.

Fig. 4-right shows the rotation error (RE) as a function of the initial CR across the 20 hard pairs. As CR increases, all methods improve. However, baseline methods only exhibit an RE reduction of about 10° for each CR step of 10°, suggesting the improvements merely echo the CR, rather than effective optimization. Thus, even at $CR = 30°$, they have over 40° residual error (Appendix Table A3). In contrast, LOTUS-based methods outperform the baselines across all CR levels, and achieve consistently good registration at CR=30°.

## 3.3. Discussion and Conclusion

We proposed LOTUS, a latent outpainting diffusion model that expands sector-shaped FOVs into rectangular ones, significantly enhancing subsequent registration performance. LOTUS not only generates realistic content but also dramatically improves inference speed and reduces computational costs compared to image-space approaches. Our results demonstrate that FOV outpainting substantially enhances 3DUS registration performance.

Our experiments only used conventional registration methods. Future work will focus on also validating LOTUS across learning-based registration approaches, as well as extending its applications to other medical imaging domains.

## Acknowledgments

This work is supported, in part, by NIH R01-HD109739, R01-HL156034, T32-EB021937, and the Vanderbilt Advanced Computing Center for Research and Education.

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

## Appendix A. Outpainting results for varying patch sizes

Fig. A1 presents the outpainting results conditioned on different patch sizes. We note that even for small patches, LOTUS is able to effectively outpaint the FOV.

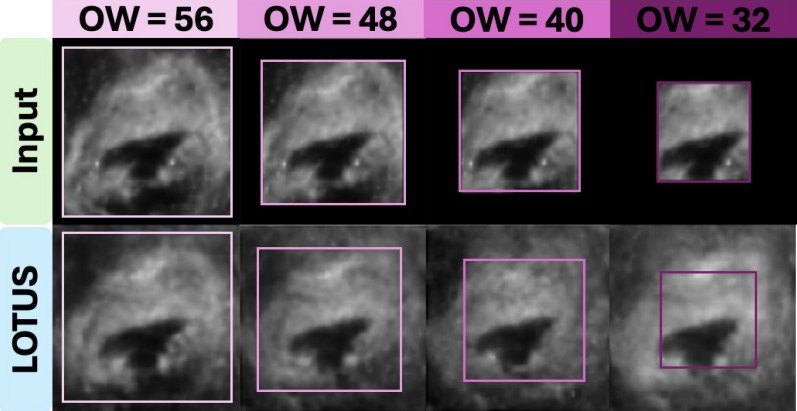

Figure A1: Outpainting performance of LOTUS under varying mask sizes. OW denotes the outpainting window size, which decreases from left to right, representing increasingly challenging outpainting tasks. LOTUS effectively removes FOV edge-related artifacts and produces realistic outpainting across different condition sizes.

## Appendix B. Additional metrics for registration performance in typical cases

Tables A1 and A2 show additional metrics for quantitative registration evaluation, complementary to Table 2 in the main manuscript. In Table A1, the Dice coefficient score (DSC) between the foreground masks of the registration pair is used to quantify the overlapping FOV area. This is to control for trivial local minima that minimize FOV overlap instead of optimizing the anatomical similarity. The moving image, despite having the largest overlap, exhibits the poorest similarity, serving as the lower bound. None of the algorithms present an alarmingly low Dice score, which shows the trivial local minima is avoided. The SSIM and PSNR metrics show similar patterns to the NCC and MSE metrics reported in Table 2. Similarly, translation error (TE) performance is consistent with the rotation error (RE) in Table 2.

## Appendix C. Quantitative registration results in hard cases

Table A3 presents the quantitative registration results for the hard cases. These correspond to the plots shown in the right panel of Fig. 4 in the main manuscript for rotation errors (RE). We also present translation errors (TE) in this Table for completeness.

Table A1: Registration performance on typical cases, using the fixed image for evaluation. **Best** and second performances are highlighted. LOTUS outpainting significantly improves performance (*, paired t-tests, p<0.05) compared to the baselines.

| Method | MOV | GT | Greedy | ANTs | LOTUS(Greedy) | LOTUS(ANTs) |
|---|---|---|---|---|---|---|
| SSIM ↑ | 0.265±0.076 | 0.513±0.070[+] | 0.354±0.115 | 0.351±0.125 | 0.490±0.102* | **0.531±0.089*** |
| PSNR ↑ | 15.1±1.4 | 19.1±1.2[+] | 16.8±2.2 | 16.7±2.1 | 19.3±3.0* | **19.3±1.2*** |
| DSC ↑ | 0.975±0.050 | 0.855±0.030 | 0.858±0.204 | 0.906±0.052 | 0.794±0.186 | 0.843±0.033 |

Table A2: Registration performance on typical cases, using the GT for evaluation. **Best** and second performances are highlighted. LOTUS outpainting significantly improves performance (*, paired t-tests, p<0.05) compared to the baselines.

| Method | Greedy | ANTs | LOTUS(Greedy) | LOTUS(ANTs) |
|---|---|---|---|---|
| SSIM ↑ | 0.612±0.252 | 0.565±0.226 | 0.746±0.169 | **0.848±0.070*** |
| PSNR ↑ | 23.2±9.4 | 20.9±6.7 | 23.8±4.3 | **26.6±2.0*** |
| TE ↓ | 25.6±23.5 | 30.3±20.7 | 11.3±19.0* | **4.33±10.05*** |

Table A3: Registration performance on hard cases, using the GT for evaluation. CR denotes the compensation rotation angle. **Best** and second performances are highlighted. LOTUS outpainting significantly decreases the rotation error (RE) and translation error (TE) (*, paired t-tests, p<0.05) compared to the baselines.

| Metric | RE ↓ | | | TE ↓ | | |
|---|---|---|---|---|---|---|
| | CR=10° | CR=20° | CR=30° | CR=10° | CR=20° | CR=30° |
| ANTs | 69.5±27.8 | 57.7±26.84 | 47.3±22.6 | 44.7±18.5 | 37.4±17.5 | 30.5±14.7 |
| Greedy | 73.7±28.8 | 62.4±36.7 | 44.6±41.7 | 47.9±19.4 | 41.8±25.2 | 29.2±28.2 |
| LOTUS(ANTs) | 63.7±25.0 | 43.8±27.6 | 21.5±21.9 | 41.1±16.1 | 28.2±18.5 | 13.9±14.0 |
| LOTUS(Greedy) | **44.7±39.4** | **33.7±34.5** | **8.35±28.68** | **28.4±25.7** | **22.2±23.4** | **5.57±20.52** |

## Appendix D. Comparison to nearest neighbor outpainting

Following a reviewer suggestion, we have padded the out-of-FOV regions using the nearest within-FOV pixel values, as shown in the New Supplementary Fig. A2. However, this padding strategy introduces a Voronoi-like unrealistic content that does not accurately reflect the underlying anatomy and therefore impact registration accuracy. We note that padding with a single constant value $c$ would not have changed the registration result compared to having the default value of $c = 0$.

**Target** **Source** **Target'** **Source'** **T' to S'** **S' to T'**

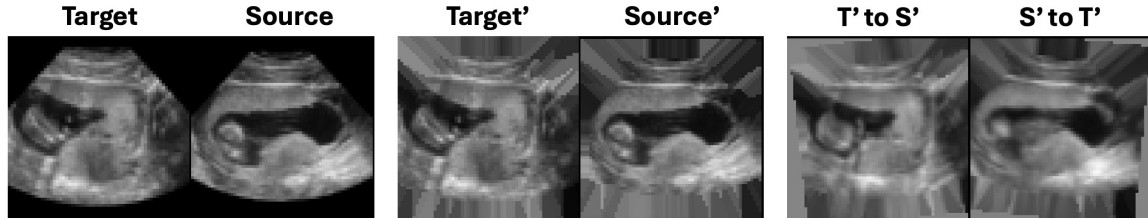

Figure A2: Padding the out-of-FOV regions of typical case 1 (from Fig. 3) the nearest within-FOV pixel value. Left: the original target and source images. Mid: padding results, Target' and Source'. Right: registration results by registering Target' to Source' and Source' to Target'.

