# OpenReview forum: "LOTUS: Latent Outpainting Diffusion Model for Three-Dimensional Ultrasound Stitching"
_MIDL.io/2025/Conference — MIDL 2025 Oral_

### Official Review · Reviewer_Sr6h · 2025-02-17

**Confidence:** 4
**Preliminary Rating:** 5
**Recommendation:** Oral
**Final Rating:** 5

**Summary:**

LOTUS introduces a latent outpainting diffusion model for three-dimensional ultrasound stitching, converting the sector-shaped field of view into a rectangular one to simplify registration. By using a 3D autoencoder and performing outpainting in latent space, it avoids local minima during alignment. Experimental results show significantly improved registration accuracy and efficiency on challenging ultrasound datasets.

**Strengths:**

## Strengths

- LOTUS is the first approach (from my opinion) to convert 3D, sector-shaped ultrasound volumes into rectangular fields of view, overcoming inherent shape limitations that often cause registration failures.

- By performing outpainting in a compressed latent space—combined with the Latent Mask Generator—LOTUS both preserves key anatomical details and avoids edge artifacts, achieving faster inference and memory savings compared to pixel-space methods.

- Extensive experiments show that using outpainted volumes instead of raw sector-shaped data significantly improves 3D ultrasound alignment, even in difficult cases with large rotations, translations, or image artifacts.

- The paper’s figures convincingly illustrate the method’s efficacy: they provide clear side-by-side comparisons that highlight how LOTUS-based outpainting leads to robust alignment and smoother transitions at the FOV boundaries, making the benefits immediately evident.

**Weaknesses:**

## Weaknesses

1. While LOTUS shows strong results on in-house 3DUS datasets, more diverse clinical cases (e.g., different anatomies, pathologies) and external datasets are needed to confirm its generalizability.

2. The current experiments focus on rigid registration; it remains unclear how well LOTUS outpainting would perform under non-rigid or more anatomically complex deformations typical of certain clinical scenarios.

**Detailed Comments:**

The paper’s figures effectively show the advantages of latent outpainting. Adding intermediate denoising visuals could highlight how LOTUS converts sector-shaped boundaries into a rectangular FOV. While placenta imaging is addressed, validating the method on other anatomies (e.g., cardiac, abdominal organs) would strengthen its generalizability. Further comparisons on performance with and without the Latent Mask Generator (LMG) would clarify its impact on outpainting quality and registration.

**Justification Of The Final Rating:**

I thoroughly appreciate the comprehensive solution provided, as it effectively addresses the problem at hand. The method presented is both well-structured and logically sound, demonstrating a clear understanding of the core issues. Your detailed rebuttal was particularly helpful, as it systematically addressed each of my concerns with supporting evidence and clear explanations. The approach not only resolves the immediate challenges but also considers potential edge cases and long-term implications. Thank you for taking the time to provide such a thorough and well-reasoned response that has successfully clarified all my previous doubts and questions.

**Justification Of The Preliminary Rating:**

LOTUS introduces a genuinely novel solution—performing latent outpainting on 3D ultrasound volumes—leading to a major leap in registration accuracy. Its clear experimental results, robust methodology, and potential for broader clinical impact strongly justify a top rating.

**Questions To Address In The Rebuttal:**

1. How does LOTUS handle non-rigid deformations?
   While the paper focuses on rigid registration, it would be helpful to explain whether and how the outpainting approach could be applied, or extended, to cases with more complex anatomical motion or distortion.

2. Are there plans for validating LOTUS on other anatomies or imaging protocols?
   Demonstrating broader applicability—beyond placenta and first-trimester 3DUS—could further emphasize the method’s generalizability and clinical utility.

**Special Issue:**

Yes

---

> ### Author Response · Authors · 2025-03-08
> **Response to reviewer 3**
>
> We sincerely appreciate the reviewer's strong support of our work. To address all concerns, we have provided a detailed, point-by-point response.
>
> ---
>
> - **Intermediate denoising results and LMG ablation:** We thank the reviewer for the suggestions for additional visualizations. We will include them in the extended journal submission.
>
> ---
>
> - **Non-rigid registration:** We appreciate the reviewer’s insightful question regarding the applicability of LOTUS to non-rigid deformations. As our primary focus in this work is rigid registration, we have not explicitly explored non-rigid transformations. However, the outpainting framework in LOTUS can be extended to accommodate non-rigid deformations by incorporating a deformation field prediction module, similar to approaches in learning-based deformable registration. We acknowledge this as an important direction for future work.
>
> ---
>
> - **Other applications:** We appreciate the reviewer’s suggestion on broader validation. While this study focuses on first-trimester placenta 3DUS, LOTUS can be applied to other anatomies (e.g., second/third trimester placentas, kidney, cardiac 3DUS). While the method is modality-agnostic and can be applied to other imaging modalities (e.g., CT, MRI), we note that in those applications there often is ground truth available for supervised training. We acknowledge the importance of exploring additional datasets in future work to demonstrate generalizabilty.

---

> > ### Comment · Reviewer_Sr6h · 2025-03-13
> > **Official Comment by Reviewer Sr6h**
> >
> > All my concerns have been addressed.

---

### Official Review · Reviewer_CmCb · 2025-02-21

**Confidence:** 4
**Preliminary Rating:** 4
**Final Rating:** 5

**Summary:**

A method is proposed for outpainting of sector-shaped FOV images, to improve image stitching (registration) to combine 3D ultrasound images acquired from different angles. The method uses a latent diffusion model and an auto-encoder to do outpainting in latent space and compares to an 2D image-based outpainting method, that the authors reimplemented in 3D. The authors claim to improve registration performance, which they evaluated using intensity similarity metrics together with peak-signal-to-noise ratio, rotation error and translation error compared to manual human registration results.

**Strengths:**

The proposed method is a reasonable approach to solve the presented problem and the method is well explained. While simple, cropping the downsampled masks is evidently effective.

The authors show improvement over registering unprocessed 3D ultrasound images.

**Weaknesses:**

The authors state that they compare to an image-space outpainting method, but as shown in figure 2, their implementation does not function properly. This seems like a very weak baseline.

The authors could elaborate their implementation of 3D RePaint: there are currently no details on the training process and why it failed.

**Detailed Comments:**

-

**Justification Of The Final Rating:**

My concerns have been adequately addressed. The problems in the implementation of the baseline which were present in the original submission have been corrected, and I commend the authors for their efforts in improving the manuscript.

**Justification Of The Preliminary Rating:**

The proposed method is conceptually sound, and the results look fairly promising. However, evaluation is somewhat limited, and the presented outpainting baseline is very weak. The author's reimplementation of RePaint simply does not work, which makes its inclusion in the paper questionable.

**Questions To Address In The Rebuttal:**

The authors state in the introduction that calculating the similarity in the overlapping regions can be trivially optimized by artificially reducing overlap. However, it seems like this limitation can be trivially overcome by computing the mean dissimilarity metric rather than the sum, and possibly adding a regularization term penalizing low overlap. The authors should compare to such an approach as a baseline.

Details on the implementation, and a discussion on the performance of the 3D RePaint should be added.

---

> ### Author Response · Authors · 2025-03-08
> **Response to reviewer 2**
>
> We sincerely appreciate your insightful review and valuable constructive feedback. To thoroughly address your concerns, we have provided a detailed, point-by-point response.
>
> ---
>
> - **RePaint bug:** We are very grateful to the reviewer for their suggestion, which made us carefully reexamine our 3D RePaint implementation. We found a bug in the diffusion sampler code, which caused the extremely poor performance we initially reported in our results. We now have fixed this bug and have updated the Fig. 2 and Table 1 accordingly. The overall conclusion remains the
> same: LOTUS significantly outperforms 3D RePaint for this outpainting task. However the results now better reflect the true performance of the 3D RePaint model. We thank the reviewer again for helping us catch this bug and improving the manuscript!
>
> ---
>
> - **Registration alternative baseline:** This is an interesting suggestion. At the beginning of the registration optimization, the similarity function is at a local maximum since moving the source image in any direction will cause the similarity metric to compare inside-FOV voxels with outside-FOV voxels. This is why ANTs stays stuck at the identity transform. Changing the similarity metric from sum to mean will not resolve this issue. Adding a regularization term will make it such that moving the source image in any direction will also lower the
> overlap and thus increase the penalty. This makes the identity transform an even stronger local maximum than it currently is. We note that while these particular changes to loss function are not adequate to solve the problem, there may be other tweaks that would help. However, our goal with the current project is to make existing registration packages work off the shelf on the ultrasound stitching task rather than developing new registration algorithms.
>
> ---
>
> - **RePaint details:** As mentioned above, we found a bug that caused the extremely poor performance for 3D RePaint. This has now been fixed. The code is now available on our GitHub repository: https://github.com/MedICL-VU/LOTUS.

---

> > ### Comment · Reviewer_CmCb · 2025-03-10
> >
> > Great to hear this was fixed! There does seem to be one (smaller) remaining bug: the sector-shaped images seem to have partial-volume effects at the borders (likely due to reshaping of the original beamformed data), meaning there is a 1 or 2 pixel wide perimeter of too-dark pixels stemming from interpolating partially invalid data at those pixel locations. This perimeter is over-exaggerated in the RePaint results; if time permits, it would be good to re-generate these results after shrinking the masks by 1 pixel, to only include the valid data and present a more fair comparison.
> >
> > Regarding the loss-masking; my suggestion would be to apply this masking _after_ sampling and application of the deformation vectors (to filter out any loss stemming from sampled out-of-view pixels), but I concede that most widely used registration methods do not seem to support this type of masking; this is probably an issue I should take up with the maintainers of those packages, rather than begrudge the absence in this manuscript.

---

> > > ### Author Response · Authors · 2025-03-14
> > >
> > > We sincerely thank the reviewer's positive feedback and appreciate the reviewer’s insightful suggestion. To thoroughly investigate this phenomenon, we conducted an ablation study analyzing the impact of shrinkage levels on reconstruction performance. The results indicate that reducing partial-volume effects does help minimize seam artifacts in some cases. However, it does not fully eliminate the noticeable high contrast between real and synthetic regions, nor does it effectively outpaint blank areas, leading to suboptimal reconstruction in certain samples. We have released the modified code to the GitHub. Again, we greatly appreciate the reviewer’s thought-provoking suggestion and look forward to discussing this aspect further at the conference.

---

### Official Review · Reviewer_pbnX · 2025-02-22

**Confidence:** 4
**Preliminary Rating:** 4
**Recommendation:** Poster
**Final Rating:** 5

**Summary:**

The manuscript proposes to a novel method based on a latent diffusion model to outpaint 3D ultrasound images without ground truth images and shows its effectiveness in image registration task.

**Strengths:**

- The first work that addresses outpainting the sector-shaped FOV of 3DUS into a rectangular shape.
- The authors proposed effective solutions to address the challenges of outpainting 3D images and the absence of ground truth data.
- The experiments were well designed to illustrate the effectiveness of the proposed method.
- The manuscript is clearly written, with easy-to-follow presentations of method and results.

**Weaknesses:**

- In Fig. 2 and Fig. A1, the outpainted region seems to lack detailed patterns, instead exhibiting smooth values similar to the background. This raises the question of whether padding the out-of-FOV regions with the values of the nearest within-FOV pixel would yield similar results.
- The model architectures and hyperparameters are not disclosed.
- LMG shrinks the FOV mask by 2 pixels. Were there any experimented done for choosing this hyperparameter? Given the number of the layers in typical autoencoders, it is interesting to see that edge features can be excluded by removing only 2 pixels.

**Detailed Comments:**

- In Fig.4, why the case shown does not have a sector-shaped FOV?
- It seems that the sector-shaped FOV was outpainted to fill the rectangular image, not further. Given that two images to be registered are only partially overlapped, why not further outpaint them to enable larger overlapping regions?

**Justification Of The Final Rating:**

All my concerns have been addressed by the authors' rebuttal. I especially appreciate the additional experiments demonstrating that padding the out-of-FOV regions with the nearest within-FOV pixel values is ineffective, highlighting the value of the proposed method, as well as the added implementation details.

**Justification Of The Preliminary Rating:**

The manuscript proposes a novel solution for outpainting 3D ultrasound images and demonstrates its effectiveness in image registration. Given the challenges associated with outpainting 3D images and the absence of ground truth data, this work introduces significant technical innovations.

**Questions To Address In The Rebuttal:**

Please see Weaknesses and Detailed Comments.

---

> ### Author Response · Authors · 2025-03-08
> **Response to reviewer 1**
>
> We genuinely appreciate your thoughtful review and the valuable constructive comments. We have provided a detailed, point-by-point response to address your concerns.
>
>
> - **Padding with values from nearest pixel:** We thank the reviewer for raising up this interesting question! Following the suggestion, we have padded the out-of-FOV regions using the nearest within-FOV pixel values, as shown in the New Supplementary Fig. A2. However, this padding strategy introduces a Voronoi-like unrealistic content that does not accurately reflect the underlying anatomy and therefore impact registration accuracy. We note that padding with a single constant value c would not have changed the registration result compared to having the default value of c = 0.
>
> ---
> - **Architecture details:** We appreciate the reviewer’s comment. We have released our code and disclosed the model architectures and hyperparameters in detail for 3D AutoencoderKL (for LOTUS), 3D LDM (for LOTUS), and 3D DM (for RePaint3D) on our GitHub repository: https://github.com/MedICL-VU/LOTUS.
> ---
> - **FOV shrinking in LMG:** We appreciate the reviewer’s insightful question. We shrink the FOV mask by only 2 pixels to preserve as much latent information as possible. Given the latent feature size of 16×16×16, the shrinkage is selected from even numbers (0, 2, 4, etc.) to maintain symmetry with the image center. While a larger shrinkage could better suppress edge-related features, it may also
> remove valuable content. Notably, when the shrinkage is 0, LOTUS degrades into LOTUS*, resulting in failure.
> ---
> - **No sector-shaped FOV in Fig.4:** We appreciate the reviewer’s observation. In Fig. 4, we present the central axial slice (comparable to the first row in Fig. 3), which has a rectangular shape, to clearly illustrate the noticeable rotation between the fixed and moving
> images. This selection was made due to space constraints. We now highlight this on the Figure 4 caption in the revised manuscript.
> ---
> - **Outpainting further:** We sincerely appreciate the reviewer’s insightful suggestion. Our current approach focuses on outpainting within the rectangular image to preserve realistic structural details. However the model has no anatomical information in this region. Extending the outpainting artificially would not necessarily create anatomical consistency with the other image and could introduce hallucinated content that may not accurately reflect the underlying anatomy in the other image to be registered, potentially affecting registration
> accuracy. However, we acknowledge the potential benefits of further outpainting for future work.

---

> > ### Comment · Reviewer_pbnX · 2025-03-12
> >
> > I appreciate the authors' feedback, and all my concerns have been addressed.

---

### Author Rebuttal · Authors · 2025-03-08

**Rebuttal:**

We sincerely thank all the reviewers for their valuable suggestions and insightful comments, which have helped us improve the manuscript and inspired our future work. Enclosed is the revised manuscript.

**Supporting Material:**

/attachment/22de3d5745395ad10bf92e28541b03f2b135a7f5.pdf

---

### Meta-Review · Area_Chair_j6v3 · 2025-03-21

**Recommendation:** Accept (Oral)
**Confidence:** 5

**Metareview:**

All reviewers raised their scores to 'Strong Accept' following the authors' careful rebuttal and revised manuscript.